# A Closer Look at Deep Policy Gradients

Andrew Ilyas[*], Logan Engstrom[*], Shibani Santurkar[1], Dimitris Tsipras[1],
Firdaus Janoos[2], Larry Rudolph[1,2], and Aleksander Mądry[1]

[1]MIT    [2]Two Sigma
{ailyas,engstrom,shibani,tsipras,madry}@mit.edu
rudolph@csail.mit.edu, firdaus.janoos@twosigma.com

## Abstract

We study how the behavior of deep policy gradient algorithms reflects the conceptual framework motivating their development. To this end, we propose a fine-grained analysis of state-of-the-art methods based on key elements of this framework: gradient estimation, value prediction, and optimization landscapes. Our results show that the behavior of deep policy gradient algorithms often deviates from what their motivating framework would predict: the surrogate objective does not match the true reward landscape, learned value estimators fail to fit the true value function, and gradient estimates poorly correlate with the "true" gradient. The mismatch between predicted and empirical behavior we uncover highlights our poor understanding of current methods, and indicates the need to move beyond current benchmark-centric evaluation methods.

## 1 Introduction

Deep reinforcement learning (RL) is behind some of the most publicized achievements of modern machine learning (Silver et al., 2017; OpenAI, 2018; Dayarathna et al., 2016; OpenAI et al., 2018). In fact, to many, this framework embodies the promise of the real-world impact of machine learning. However, the deep RL toolkit has not yet attained the same level of engineering stability as, for example, the current deep (supervised) learning framework. Indeed, recent studies demonstrate that state-of-the-art deep RL algorithms suffer from oversensitivity to hyperparameter choices, lack of consistency, and poor reproducibility (Henderson et al., 2017).

This state of affairs suggests that it might be necessary to re-examine the conceptual underpinnings of deep RL methodology. More precisely, the overarching question that motivates this work is:

*To what degree does current practice in deep RL reflect the principles informing its development?*

Our specific focus is on deep policy gradient methods, a widely used class of deep RL algorithms. Our goal is to explore the extent to which state-of-the-art implementations of these methods succeed at realizing the key primitives of the general policy gradient framework.

**Our contributions.** We take a broader look at policy gradient algorithms and their relation to their underlying framework. With this perspective in mind, we perform a fine-grained examination of key RL primitives as they manifest in practice. Concretely, we study:

- **Gradient Estimation:** we find that even when agents improve in reward, their gradient estimates used in parameter updates poorly correlate with the "true" gradient. We additionally show that gradient estimate quality decays with training progress and task complexity. Finally, we demonstrate that varying the sample regime yields training dynamics that are unexplained by the motivating framework and run contrary to supervised learning intuition.

- **Value Prediction:** our experiments indicate that value networks successfully solve the supervised learning task they are trained on, but do *not* fit the true value function. Additionally, employing a value network as a baseline function only marginally decreases the

---
[*]Equal contribution. Work done in part as an intern at Two Sigma.

variance of gradient estimates compared to using true value as a baseline (but still dramatically increases agent's performance compared to using no baseline at all).

- **Optimization Landscapes:** we show that the optimization landscape induced by modern policy gradient algorithms is often not reflective of the underlying true reward landscape, and that the latter is frequently poorly behaved in the relevant sample regime.

Overall, our results demonstrate that the motivating theoretical framework for deep RL algorithms is often unpredictive of phenomena arising in practice. This suggests that building reliable deep RL algorithms requires moving past benchmark-centric evaluations to a multi-faceted understanding of their often unintuitive behavior. We conclude (in Section 3) by discussing several areas where such understanding is most critically needed.

## 2 EXAMINING THE PRIMITIVES OF DEEP POLICY GRADIENT ALGORITHMS

In this section, we investigate the degree to which our theoretical understanding of RL applies to modern methods. We consider key primitives of policy gradient algorithms: gradient estimation, value prediction and reward fitting. In what follows, we perform a fine-grained analysis of state-of-the-art policy gradient algorithms (PPO and TRPO) through the lens of these primitives—detailed preliminaries, background, and notation can be found in Appendix A.1.

### 2.1 GRADIENT ESTIMATE QUALITY

A central premise of policy gradient methods is that stochastic gradient ascent on a suitable objective function yields a good policy. These algorithms use as a primitive the gradient of that objective function:

$$\hat{g} = \nabla_\theta \mathbb{E}_{(s_t,a_t)\sim\pi_0}\left[\frac{\pi_\theta(a_t|s_t)}{\pi_0(a_t|s_t)}\widehat{A}_{\pi_0}(s_t,a_t)\right] = \mathbb{E}_{(s_t,a_t)\sim\pi_0}\left[\frac{\nabla_\theta\pi_\theta(a_t|s_t)}{\pi_0(a_t|s_t)}\widehat{A}_{\pi_0}(s_t,a_t)\right], \quad (1)$$

where in the above we use standard RL notation (see Appendix A.1 for more details). An underlying assumption behind these methods is that we have access to a reasonable estimate of this quantity. This assumption effectively translates into an assumption that we can accurately estimate the expectation above using an empirical mean of finite (typically $\sim 10^3$) samples. Evidently (since the agent attains a high reward) these estimates are sufficient to consistently improve reward—we are thus interested in the relative quality of these gradient estimates in practice, and the effect of gradient quality on optimization.

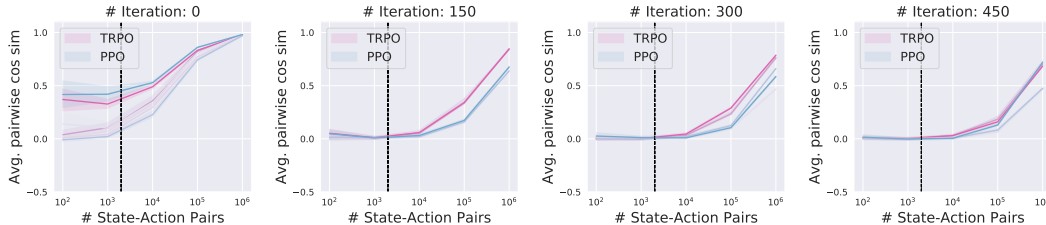

Figure 1: Empirical variance of the estimated gradient (c.f. (1)) as a function of the number of state-action pairs used in estimation in the MuJoCo Humanoid task. We measure the average pairwise cosine similarity between ten repeated gradient measurements taken from the same policy, with the 95% confidence intervals (shaded). For each algorithm, we perform multiple trials with the same hyperparameter configurations but different random seeds, shown as repeated lines in the figure. The vertical line (at $x = 2$K) indicates the sample regime used for gradient estimation in standard implementations of policy gradient methods. In general, it seems that obtaining tightly concentrated gradient estimates would require significantly more samples than are used in practice, particularly after the first few timesteps. For other tasks – such as Walker2d-v2 and Hopper-v2 – the plots (seen in Appendix Figure 9) have similar trends, except that gradient variance is slightly lower. Confidence intervals calculated with 500 sample bootstrapping.

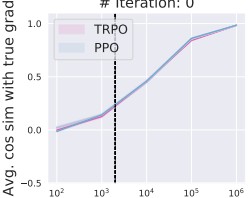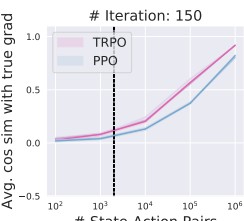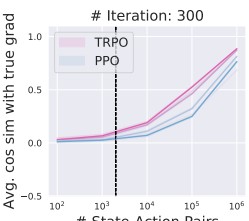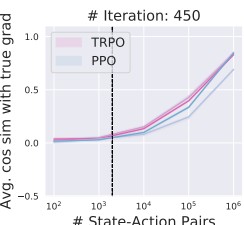

Figure 2: Convergence of gradient estimates (c.f. (1)) to the "true" expected gradient in the MuJoCo Humanoid task. We measure the mean cosine similarity between the "true" gradient approximated using ten million state-action pairs, and ten gradient estimates which use increasing numbers of state-action pairs (with 95% confidence intervals). For each algorithm, we perform multiple trials with the same hyperparameter configurations but different random seeds. The vertical line (at $x = 2K$) indicates the sample regime used for gradient estimation in standard implementations of policy gradient methods. Observe that although it is possible to empirically estimate the true gradient, this requires several-fold more samples than are used commonly in practical applications of these algorithms. See additionally that the estimation task becomes more difficult further into training. For other tasks – such as Walker2d-v2 and Hopper-v2 – the plots (seen in Appendix Figure 10) have similar trends, except that gradient estimation is slightly better. Confidence intervals calculated with 500 sample bootstrapping.

**How accurate are the gradient estimates we compute?** To answer this question, we examine two of the most natural measures of estimate quality: the empirical variance and the convergence to the "true" gradient. To evaluate the former, we measure the average pairwise cosine similarity between estimates of the gradient computed from the same policy with independent rollouts (Figure 1). We evaluate the latter by first forming an estimate of the true gradient with a large number of state-action pairs. We then examine the convergence of gradient estimates to this "true" gradient (which we once again measure using cosine similarity) as we increase the number of samples (Figure 2).

We observe that *deep policy gradient methods operate with relatively poor estimates of the gradient*, especially as task complexity increases and as training progresses (contrast Humanoid-v2, a "hard" task, to other tasks and contrast successive checkpoints in Figures 1 and 2). This is in spite of the fact that our agents continually improve throughout training, and attain nowhere near the maximum reward possible on each task. In fact, we sometimes observe a *zero* or *even negative* correlation in the relevant sample regime[1].

While these results might be reminiscent of the well-studied "noisy gradients" problem in supervised learning (Robbins & Monro, 1951; d'Aspremont, 2008; Kawaguchi, 2016; Safran & Shamir, 2018; Livni et al., 2014; Keskar et al., 2016; Hochreiter & Schmidhuber, 1997), we have very little understanding of how gradient quality affects optimization in the substantially different reinforcement learning setting. For example:

- The sample regime in which RL algorithms operate seems to have a profound impact on the *robustness and stability of agent training*—in particular, many of the sensitivity issues reported by Henderson et al. (2017) are claimed to disappear (Sutskever, 2018) in higher-sample regimes. Understanding the implications of working in this sample regime, and more generally the impact of sample complexity on training stability remains to be precisely understood.

- Agent policy networks are trained concurrently with *value networks* (discussed more in the following section) meant to reduce the variance of gradient estimates. Under our conceptual framework, we might expect these networks to help gradient estimates more as training progresses, contrary to what we observe in Figure 1. The value network also makes the now *two-player* optimization landscape and training dynamics even more difficult to grasp, as such interactions are poorly understood.

---

[1]Deep policy gradient algorithms use gradients indirectly to compute steps—in Appendix A.4 we show that our results also hold true for these computed steps.

- The relevant measure of sample complexity for many settings (number of state-action pairs) can differ drastically from the number of *independent* samples used at each training iteration (the number of complete trajectories). The latter quantity (a) tends to be much lower than the number of state-action pairs, and (b) decreases across iterations during training.

All the above factors make it unclear to what degree our intuition from classical settings transfer to the deep RL regime. And the policy gradient framework, as of now, provides little predictive power regarding the variance of gradient estimates and its impact on reward optimization.

Our results indicate that despite having a rigorous theoretical framework for RL, we lack a precise understanding of the structure of the reward landscape and optimization process.

## 2.2 VALUE PREDICTION

Our findings from the previous section motivate a deeper look into gradient estimation. After all, the policy gradient in its original formulation (Sutton et al., 1999) is known to be hard to estimate, and thus algorithms employ a variety of variance reduction methods. The most popular of these techniques is a baseline function. Concretely, an equivalent form of the policy gradient is given by:

$$\widehat{g}_\theta = \mathbb{E}_{\tau \sim \pi_\theta} \left[ \sum_{(s_t, a_t) \in \tau} \nabla_\theta \log \pi_\theta(a_t|s_t) \cdot (Q_{\pi_\theta}(s_t, a_t) - b(s_t)) \right] \tag{2}$$

where $b(s_t)$ is some fixed function of the state $s_t$. A canonical choice of baseline function is the value function $V_\pi(s)$, the expected return from a given state (more details and motivation in A.1):

$$V_{\pi_\theta}(s_t) = \mathbb{E}_{\pi_\theta}[R_t|s_t] . \tag{3}$$

Indeed, fitting a value-estimating function (Schulman et al., 2015c; Sutton & Barto, 2018) (a neural network, in the deep RL setting) and using it as a baseline function is precisely the approach taken by most deep policy gradient methods. Concretely, one trains a value network $V_{\theta_t}^\pi$ such that:

$$\theta_t = \min_\theta \mathbb{E} \left[ \left( V_\theta^\pi(s_t) - (V_{\theta_{t-1}}^\pi(s_t) + A_t) \right)^2 \right] \tag{4}$$

where $V_{\theta_{t-1}}^\pi(s_t)$ are estimates given by the last value function, and $A_t$ is the *advantage* of the policy, i.e. the returns minus the estimated values. (Typically, $A_t$ is estimated using generalized advantage estimation, as described in (Schulman et al., 2015c).) Our findings in the previous section prompt us to take a closer look at the value network and its impact on the variance of gradient estimates.

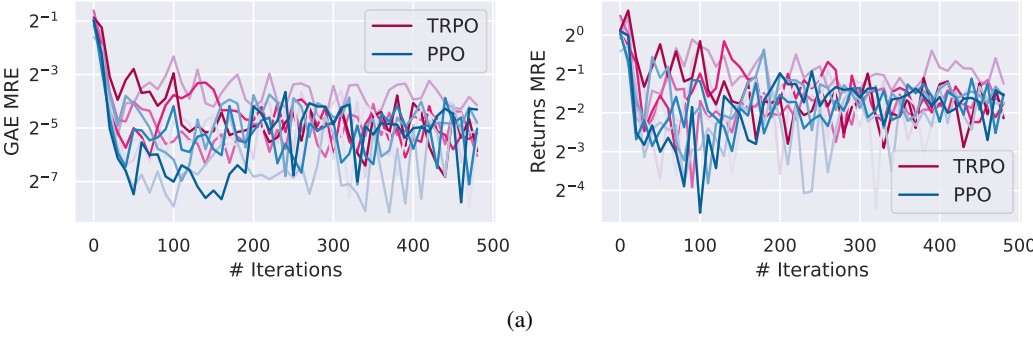

(a)

Figure 3: Quality of value prediction in terms of mean relative error (MRE) on heldout state-action pairs for agents trained to solve the MuJoCo Walker2d-v2 task. We observe in (*left*) that the agents do indeed succeed at solving the supervised learning task they are trained for—the MRE on the GAE-based value loss $(V_{old} + A_{GAE})^2$ (c.f. (4)) is small. On the other hand, in (*right*) we see that the returns MRE is still quite high—the learned value function is off by about $50\%$ with respect to the underlying true value function. Similar plots for other MuJoCo tasks are in Appendix A.5.

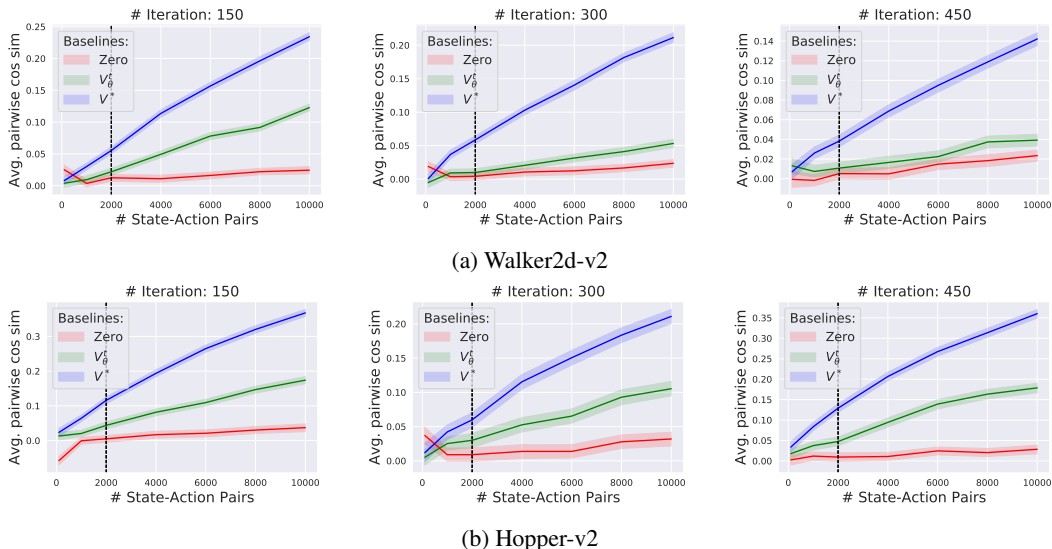

Figure 4: Efficacy of the value network as a variance reducing baseline for Walker2d-v2 (top) and Hopper-v2 (bottom) agents. We measure the empirical variance of the gradient (c.f. (1)) as a function of the number of state-action pairs used in estimation, for different choices of baseline functions: the value network (used by the agent in training), the "true" value function (fit to the returns using $5 \cdot 10^6$ state-action pairs sampled from the *current* policy) and the "zero" value function (i.e. replacing advantages with returns). We observe that using the true value function leads to a significantly lower-variance estimate of the gradient compared to the value network. In turn, employing the value network yields a noticeable variance reduction compared to the zero baseline function, even though this difference may appear rather small in the small-sample regime (2K). Confidence intervals calculated with 10 sample bootstrapping.

**Value prediction as a supervised learning problem.** We first analyze the value network through the lens of the supervised learning problem it solves. After all, (4) describes an empirical risk minimization, where a loss is minimized over a set of sampled $(s_t, a_t)$. So, how does $V_\theta^\pi$ perform as a solution to (4)? And in turn, how does (4) perform as a proxy for learning the true value function?

Our results (Figure 3a) show that the value network *does* succeed at both fitting the given loss function and generalizing to unseen data, showing low and stable mean relative error (MRE). However, the significant drop in performance as shown in Figure 3 indicates that the supervised learning problem induced by (4) *does not* lead to $V_\theta^\pi$ learning the underlying true value function.

**Does the value network lead to a reduction in variance?** Though evaluating the $V_\theta^\pi$ baseline function as a value predictor as we did above is informative, in the end the sole purpose of the value function is to reduce variance. So: how does using our value function actually impact the variance of our gradient estimates? To answer this question, we compare the variance reduction that results from employing our value network against both a "true" value function and a trivial "zero" baseline function (i.e. simply replacing advantages with returns). Our results, captured in Figure 4, show that the "true" value function yields a much lower-variance estimate of the gradient. This is especially true in the sample regime in which we operate. We note, however, that despite not effectively predicting the true value function or inducing the same degree of variance reduction, the value network *does* help to some degree (compared to the "zero" baseline). Additionally, the seemingly marginal increase in gradient correlation provided by the value network (compared to the "true" baseline function) turns out to result in a significant improvement in agent performance. (Indeed, agents trained without a baseline reach almost an order of magnitude worse reward.)

Our findings suggest that we still need a better understanding of the role of the value network in agent training, and raise several questions that we discuss in Section 3.

## 2.3 Exploring the optimization landscape

Another key assumption of policy gradient algorithms is that first-order updates (w.r.t. policy parameters) actually yield better policies. It is thus natural to examine how valid this assumption is.

**The true rewards landscape.** We begin by examining the landscape of agent reward with respect to the policy parameters. Indeed, even if deep policy gradient methods do not optimize for the true reward directly (e.g. if they use a surrogate objective), the ultimate goal of any policy gradient algorithm is to navigate this landscape. First, Figure 5 shows that while estimating the true reward landscape with a high number of samples yields a relatively smooth reward landscape (perhaps suggesting viability of direct reward optimization), estimating the true reward landscape in the typical, low sample regime results in a landscape that appears jagged and poorly-behaved. The low-sample regime thus gives rise to a certain kind of barrier to direct reward optimization. Indeed, applying our algorithms in this regime makes it impossible to distinguish between good and bad points in the landscape, even though the true underlying landscape is fairly well-behaved.

**The surrogate objective landscape.** The untamed nature of the rewards landscape has led to the development of alternate approaches to reward maximization. Recall that an important element of many modern policy gradient methods is the maximization of a surrogate objective function in place of the true rewards (the exact mechanism behind the surrogate objective is detailed in Appendix A.1, and particularly in (14)). The surrogate objective, based on relaxing the policy improvement theorem of Kakade and Langford (Kakade & Langford, 2002), can be viewed as a simplification of the reward maximization objective.

As a purported approximation of the true returns, one would expect that the surrogate objective landscape approximates the true reward landscape fairly well. That is, parameters corresponding to good surrogate objective will also correspond to good true reward.

Figure 6 shows that in the early stages of training, the optimization landscapes of the true reward and surrogate objective are indeed approximately aligned. However, as training progresses, the surrogate objective becomes much less predictive of the true reward in the relevant sample regime. In particular, we often observe that directions that *increase* the surrogate objective lead to a *decrease* of the true reward (see Figures 6, 7). In a higher-sample regime (using several orders of magnitude more samples), we find that PPO and TRPO turn out to behave rather differently. In the case of TRPO, the update direction following the surrogate objective matches the true reward much more closely. However, for PPO we consistently observe landscapes where the step direction leads to lower true reward, even in the high-sample regime. This suggests that even when estimated accurately enough,

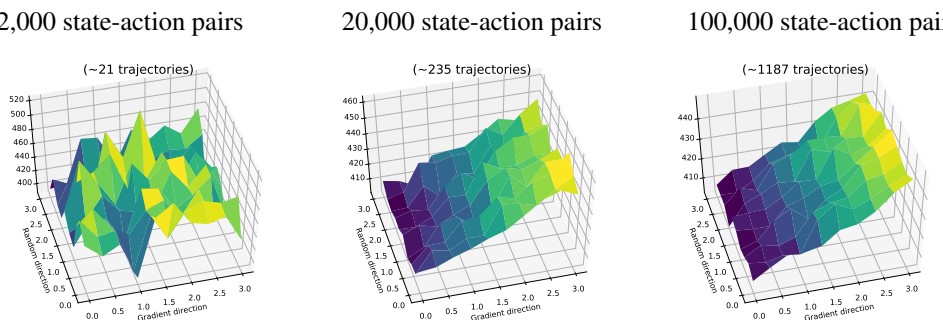

Figure 5: True reward landscape concentration for TRPO on Humanoid-v2. We visualize the landscape at a training iteration 150 while varying the number of trajectories used in reward estimation (each subplot), both in the direction of the step taken and a random direction. Moving one unit along the "step direction" axis corresponds to moving one full step in parameter space. In the random direction one unit corresponds to moving along a random norm 2 Gaussian vector in the parameter space. In practice, the norm of the step is typically an order of magnitude lower than the random direction. While the landscape is very noisy in the low-sample regime, large numbers of samples reveal a well-behaved underlying landscape. See Figures 20, 19 of the Appendix for additional plots.

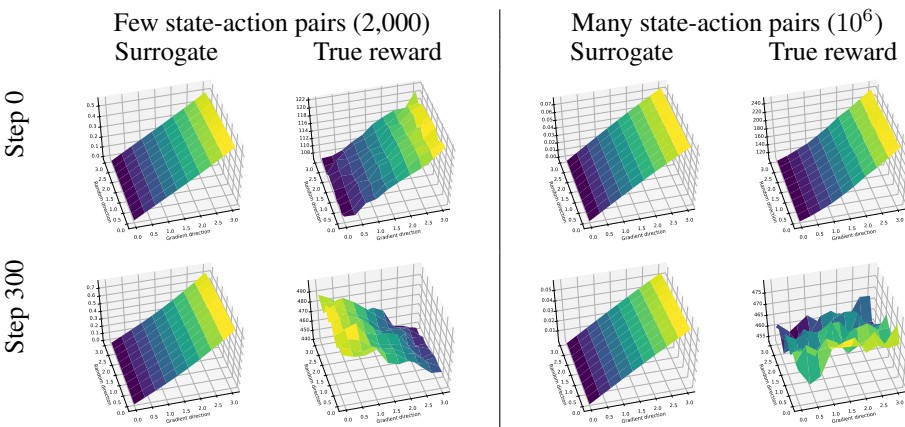

Figure 6: True reward and surrogate objective landscapes for TRPO on the Humanoid-v2 MuJoCo task. We visualize the landscapes in the direction of the update step and a random direction (as in Figure 5). The surrogate objective corresponds to the actual function optimized by the algorithm at each step. We estimate true reward with $10^6$ state-action pairs per point. We compare the landscapes at different points in training and with varying numbers of state-action pairs used in the update step. Early in training the true and surrogate landscapes align fairly well in both sample regimes, but later become misaligned in the low-sample regime. More landscapes in Appendix Figures 13-18.

the surrogate objective might not be an accurate proxy for the true reward. (Recall from Section 2.1 that this is a sample regime where we *are* able to estimate the true gradient of the reward fairly well.)

## 3    TOWARDS STRONGER FOUNDATIONS FOR DEEP RL

Deep reinforcement learning (RL) algorithms have shown great practical promise, and are rooted in a well-grounded theoretical framework. However, our results indicate that this framework often fails to provide insight into the practical performance of these algorithms. This disconnect impedes our understanding of why these algorithms succeed (or fail), and is a major barrier to addressing key challenges facing deep RL such as brittleness and poor reproducibility.

To close this gap, we need to either develop methods that adhere more closely to theory, or build theory that can capture what makes existing policy gradient methods successful. In both cases, the first step is to precisely pinpoint where theory and practice diverge. To this end, we analyze and consolidate our findings from the previous section.

**Gradient estimation.** Our analysis in Section 2.1 shows that the quality of gradient estimates that deep policy gradient algorithms use is rather poor. Indeed, even when agents improve, such gradient estimates often poorly correlate with the true gradient (c.f. Figure 2). We also note that gradient correlation decreases as training progresses and task complexity increases. While this certainly does not preclude the estimates from conveying useful signal, the exact underpinnings of this phenomenon in deep RL still elude us. In particular, in Section 2.1 we outline a few keys ways in which the deep RL setting is quite unique and difficult to understand from an optimization perspective, both theoretically and in practice Overall, understanding the impact of gradient estimate quality on deep RL algorithms is challenging and largely unexplored.

**Value prediction.** The findings presented in Section 2.2 identify two key issues. First, while the value network successfully solves the supervised learning task it is trained on, it does not accurately model the "true" value function. Second, employing the value network as a baseline does decrease the gradient variance (compared to the trivial ("zero") baseline). However, this decrease is rather marginal compared to the variance reduction offered by the "true" value function.

It is natural to wonder whether this failure in modeling the value function is inevitable. For example, how does the loss function used to train the value network impact value prediction and variance reduction? More broadly, we lack an understanding of the precise role of the value network in

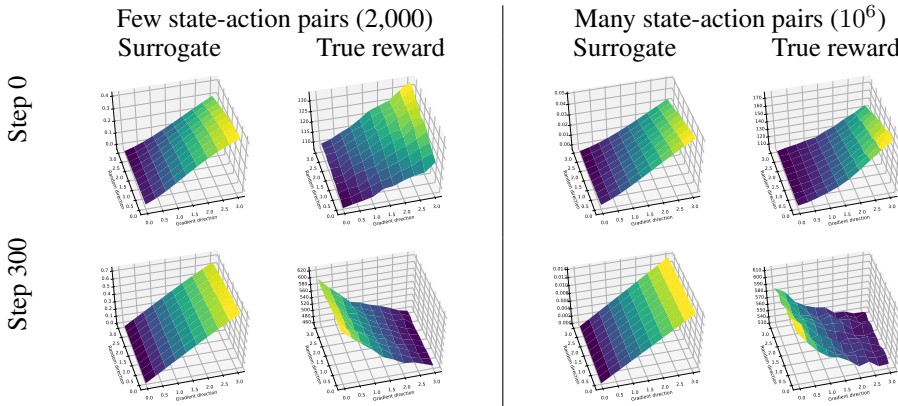

Figure 7: True reward and surrogate objective landscapes for PPO on the Humanoid-v2 MuJoCo task. See Figure 6 for a description. We observe that early in training the true and surrogate landscapes align well. However, later increasing the surrogate objective leads to lower true reward.

training. Can we empirically quantify the relationship between variance reduction and performance? And does the value network play a broader role than just variance reduction?

**Optimization landscape.** We have also seen, in Section 2.3, that the optimization landscape induced by modern policy gradient algorithms, the surrogate objective, is often not reflective of the underlying true reward landscape. We thus need a deeper understanding of why current methods succeed despite these issues, and, more broadly, how to better navigate the true reward landscape.

## 4 RELATED WORK

The idea of using gradient estimates to update neural network-based RL agents dates back at least to the REINFORCE (Williams, 1992) algorithm. Later, Sutton (Sutton et al., 1999) established a unifying framework casting these algorithms as instances of the policy gradient class of algorithms.

Our work focuses on proximal policy optimization (PPO) (Schulman et al., 2017) and trust region policy optimization (TRPO) (Schulman et al., 2015a), which are two of the most prominent policy gradient algorithms used in deep RL, drawing inspiration from works on related algorithms, such as (Peters et al., 2010) and Kakade (2001).

Many recent works document the brittleness of deep RL algorithms (Henderson et al., 2018; 2017; Islam et al., 2017). (Rajeswaran et al., 2017) and (Mania et al., 2018) demonstrate that on many benchmark tasks, state-of-the-art performance can be attained by augmented randomized search approaches. McCandlish et al. (2018) investigates gradient noise in large-batch settings, and Ahmed et al. (2018) investigates the role of *entropy regularization* (which we do not study) on optimization.

## 5 CONCLUSION

In this work, we analyze the degree to which key primitives of deep policy gradient algorithms follow their conceptual underpinnings. Our experiments show that these primitives often do not conform to the expected behavior: gradient estimates poorly correlate with the true gradient, better gradient estimates can require lower learning rates and can induce degenerate agent behavior, value networks reduce gradient estimation variance to a significantly smaller extent than the true value, and the underlying optimization landscape can be misleading.

This demonstrates that there is a significant gap between the theory inspiring current algorithms and the actual mechanisms driving their performance. Overall, our findings suggest that developing a deep RL toolkit that is truly robust and reliable will require moving beyond the current benchmark-driven evaluation model to a more fine-grained understanding of deep RL algorithms.

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

# A APPENDIX

## A.1 BACKGROUND

In the reinforcement learning (RL) setting, an agent interacts with a stateful environment with the goal of maximizing cumulative reward. Formally, we model the environment as a (possibly randomized) function mapping its current state $s$ and an action $a$ supplied by the agent to a new state $s'$ and a resulting reward $r$. The choice of actions of the agent is governed by the its *policy* $\pi$. This policy is a function mapping environment states to a distribution over the actions to take. The objective of an RL algorithm is to find a policy $\pi$ which maximizes the expected cumulative reward, where the expectation is taken over both environment randomness and the (randomized) action choices.

**Preliminaries and notation.** For a given policy $\pi$, we denote by $\pi(a|s)$ the probability that this policy assigns to taking action $a$ when the environment is in the state $s$. We use $r(s, a)$ to denote the reward that the agent earns for playing action $a$ in response to the state $s$. A *trajectory* $\tau = \{(a_t, s_t) : t \in \{1 \ldots T\}\}$ is a sequence of state-action pairs that constitutes a valid transcript of interactions of the agent with the environment. (Here, $a_t$ (resp. $s_t$) corresponds to the action taken by the agent (resp. state of the environment) in the $t$-th round of interaction.) We then define $\pi(\tau)$ to be the probability that the trajectory $\tau$ is executed if the agent follows policy $\pi$ (provided the initial state of the environment is $s_1$). Similarly, $r(\tau) = \sum_t r(s_t, a_t)$ denotes the cumulative reward earned by the agent when following this trajectory, where $s_t$ (resp. $a_t$) denote the $t$-th state (resp. action) in the trajectory $\tau$. In the RL setting, however, we often choose to maximize the *discounted* cumulative reward of a policy $R := R_1$, where $R_t$ is defined as

$$R_t(\tau) = \sum_{t'=t}^{\infty} \gamma^{(t'-t)} r_{t'} \ .$$

and $0 < \gamma < 1$ is a "discount factor". The discount factor ensures that the cumulative reward of a policy is well-defined even for an infinite time horizon, and it also incentivizes achieving reward earlier.

**Policy gradient methods.** A widely used class of RL algorithms that will be the focus of our analysis is the class of so-called *policy gradient methods*. The central idea behind these algorithms is to first parameterize the policy $\pi_\theta$ using a parameter vector $\theta$. (In the deep RL context, $\pi_\theta$ is expressed by a neural network with weights $\theta$.) Then, we perform stochastic gradient ascent on the cumulative reward with respect to $\theta$. In other words, we want to apply the stochastic ascent approach to our problem:

$$\max_\theta \mathbb{E}_{\tau \sim \pi_\theta}[r(\tau)] \ , \tag{5}$$

where $\tau \sim \pi_\theta$ represents trajectories (rollouts) sampled from the distribution induced by the policy $\pi_\theta$. This approach relies on the key observation (Sutton et al., 1999) that under mild conditions, the gradient of our objective can be written as:

$$\nabla_\theta \mathbb{E}_{\tau \sim \pi_\theta}[r(\tau)] = \mathbb{E}_{\tau \sim \pi_\theta}[\nabla_\theta \log(\pi_\theta(\tau)) \ r(\tau)], \tag{6}$$

and the latter quantity can be estimated directly by sampling trajectories according to the policy $\pi_\theta$.

When we use the discounted variant of the cumulative reward and note that the action of the policy at time $t$ cannot affect its performance at earlier times, we can express our gradient estimate as:

$$\widehat{g_\theta} = \mathbb{E}_{\tau \sim \pi_\theta} \left[ \sum_{(s_t, a_t) \in \tau} \nabla_\theta \log \pi_\theta(a_t|s_t) \cdot Q_{\pi_\theta}(s_t, a_t) \right] \ , \tag{7}$$

where $Q_{\pi_\theta}(s_t, a_t)$ represents the expected returns after taking action $a_t$ from state $s_t$:

$$Q_{\pi_\theta}(s_t, a_t) = \mathbb{E}_{\pi_\theta}[R_t|a_t, s_t] \ . \tag{8}$$

**Value estimation and advantage.** Unfortunately, the variance of the expectation in (7) can be (and often is) very large, which makes getting an accurate estimate of this expectation quite challenging. To alleviate this issue, a number of variance reduction techniques have been developed. One of the most popular such techniques is the use of a so-called baseline function, wherein a state-dependent value is subtracted from $Q_{\pi_\theta}$. Thus, instead of estimating (7) directly, we use:

$$\widehat{g}_\theta = \mathbb{E}_{\tau \sim \pi_\theta} \left[ \sum_{(s_t, a_t) \in \tau} \nabla_\theta \log \pi_\theta(a_t | s_t) \cdot (Q_{\pi_\theta}(s_t, a_t) - b(s_t)) \right], \tag{9}$$

where $b(\cdot)$ is a baseline function of our choice.

A natural choice of the baseline function is the value function, i.e.

$$V_{\pi_\theta}(s_t) = \mathbb{E}_{\pi_\theta}[R_t | s_t] . \tag{10}$$

When we use the value function as our baseline, the resulting gradient estimation problem becomes:

$$\widehat{g}_\theta = \mathbb{E}_{\tau \sim \pi_\theta} \left[ \sum_{(s_t, a_t) \in \tau} \nabla_\theta \log \pi_\theta(a_t | s_t) \cdot A_{\pi_\theta}(s_t, a_t) \right], \tag{11}$$

where

$$A_{\pi_\theta}(s_t, a_t) = Q_{\pi_\theta}(s_t, a_t) - V_{\pi_\theta}(s_t) \tag{12}$$

is referred to as the *advantage* of performing action $a_t$. Different methods of estimating $V_{\pi_\theta}$ have been proposed, with techniques ranging from moving averages to the use of neural network predictors Schulman et al. (2015b).

**Surrogate Objective.** So far, our focus has been on extracting a good estimate of the gradient with respect to the policy parameters $\theta$. However, it turns out that directly optimizing the cumulative rewards can be challenging. Thus, a modification used by modern policy gradient algorithms is to optimize a "surrogate objective" instead. We will focus on maximizing the following local approximation of the true reward Schulman et al. (2015a):

$$\max_\theta \ \mathbb{E}_{(s_t, a_t) \sim \pi} \left[ \frac{\pi_\theta(a_t | s_t)}{\pi(a_t | s_t)} A_\pi(s_t, a_t) \right] \qquad \left( = \mathbb{E}_{\pi_\theta} [A_\pi] \right), \tag{13}$$

or the normalized advantage variant proposed to reduce variance Schulman et al. (2017):

$$\max_\theta \ \mathbb{E}_{(s_t, a_t) \sim \pi} \left[ \frac{\pi_\theta(a_t | s_t)}{\pi(a_t | s_t)} \widehat{A}_\pi(s_t, a_t) \right] \tag{14}$$

where

$$\widehat{A}_\pi = \frac{A_\pi - \mu(A_\pi)}{\sigma(A_\pi)} \tag{15}$$

and $\pi$ is the current policy.

**Trust region methods.** The surrogate objective function, although easier to optimize, comes at a cost: the gradient of the surrogate objective is only predictive of the policy gradient locally (at the current policy). Thus, to ensure that our update steps we derive based on the surrogate objective are predictive, they need to be confined to a "trust region" around the current policy. The resulting trust region methods (Kakade, 2001; Schulman et al., 2015a; 2017) try to constrain the local variation of the parameters in policy-space by restricting the distributional distance between successive policies.

A popular method in this class is trust region policy optimization (TRPO) Schulman et al. (2015a), which constrains the KL divergence between successive policies on the optimization trajectory, leading to the following problem:

$$\max_\theta \ \mathbb{E}_{(s_t, a_t) \sim \pi} \left[ \frac{\pi_\theta(a_t | s_t)}{\pi(a_t | s_t)} \widehat{A}_\pi(s_t, a_t) \right]$$
$$\text{s.t.} \quad D_{KL}(\pi_\theta(\cdot \mid s) || \pi(\cdot \mid s)) \le \delta, \quad \forall s . \tag{16}$$

In practice, this objective is maximized using a second-order approximation of the KL divergence and natural gradient descent, while replacing the worst-case KL constraints over all possible states with an approximation of the mean KL based on the states observed in the current trajectory.

**Proximal policy optimization.**    In practice, the TRPO algorithm can be computationally costly—the step direction is estimated with nonlinear conjugate gradients, which requires the computation of multiple Hessian-vector products. To address this issue, Schulman et al. Schulman et al. (2017) propose proximal policy optimization (PPO), which utilizes a different objective and does not compute a projection. Concretely, PPO proposes replacing the KL-constrained objective (16) of TRPO by clipping the objective function directly as:

$$\max_{\theta} \, \mathbb{E}_{(s_t, a_t) \sim \pi} \left[ \min \left( \mathrm{clip} \left( \rho_t, 1 - \varepsilon, 1 + \varepsilon \right) \widehat{A}_{\pi}(s_t, a_t), \; \rho_t \widehat{A}_{\pi}(s_t, a_t) \right) \right] \tag{17}$$

where

$$\rho_t = \frac{\pi_{\theta}(a_t | s_t)}{\pi(a_t | s_t)} \tag{18}$$

In addition to being simpler, PPO is intended to be faster and more sample-efficient than TRPO (Schulman et al., 2017).

## A.2 Experimental Setup

We use the following parameters for PPO and TRPO based on a hyperparameter grid search:

Table 1: Hyperparameters for PPO and TRPO algorithms.

| | Humanoid-v2 | | Walker2d-v2 | | Hopper-v2 | |
|---|---|---|---|---|---|---|
| | PPO | TRPO | PPO | TRPO | PPO | TRPO |
| Timesteps per iteration | 2048 | 2048 | 2048 | 2048 | 2048 | 2048 |
| Discount factor ($\gamma$) | 0.99 | 0.99 | 0.99 | 0.99 | 0.99 | 0.99 |
| GAE discount ($\lambda$) | 0.95 | 0.95 | 0.95 | 0.95 | 0.95 | 0.95 |
| Value network LR | 0.0001 | 0.0003 | 0.0003 | 0.0003 | 0.0002 | 0.0002 |
| Value net num. epochs | 10 | 10 | 10 | 10 | 10 | 10 |
| Policy net hidden layers | [64, 64] | [64, 64] | [64, 64] | [64, 64] | [64, 64] | [64, 64] |
| Value net hidden layers | [64, 64] | [64, 64] | [64, 64] | [64, 64] | [64, 64] | [64, 64] |
| KL constraint ($\delta$) | N/A | 0.07 | N/A | 0.04 | N/A | 0.13 |
| Fisher est. fraction | N/A | 0.1 | N/A | 0.1 | N/A | 0.1 |
| Conjugate grad. steps | N/A | 10 | N/A | 10 | N/A | 10 |
| CG damping | N/A | 0.1 | N/A | 0.1 | N/A | 0.1 |
| Backtracking steps | N/A | 10 | N/A | 10 | N/A | 10 |
| Policy LR (Adam) | 0.00025 | N/A | 0.0004 | N/A | 0.00045 | N/A |
| Policy epochs | 10 | N/A | 10 | N/A | 10 | N/A |
| PPO Clipping $\varepsilon$ | 0.2 | N/A | 0.2 | N/A | 0.2 | N/A |
| Entropy coeff. | 0.0 | 0.0 | 0.0 | 0.0 | 0.0 | 0.0 |
| Reward clipping | [-10, 10] | – | [-10, 10] | – | [-10, 10] | – |
| Reward normalization | On | Off | On | Off | On | Off |
| State clipping | [-10, 10] | – | [-10, 10] | – | [-10, 10] | – |

All error bars we plot are 95% confidence intervals, obtained via bootstrapped sampling.

## A.3 STANDARD REWARD PLOTS

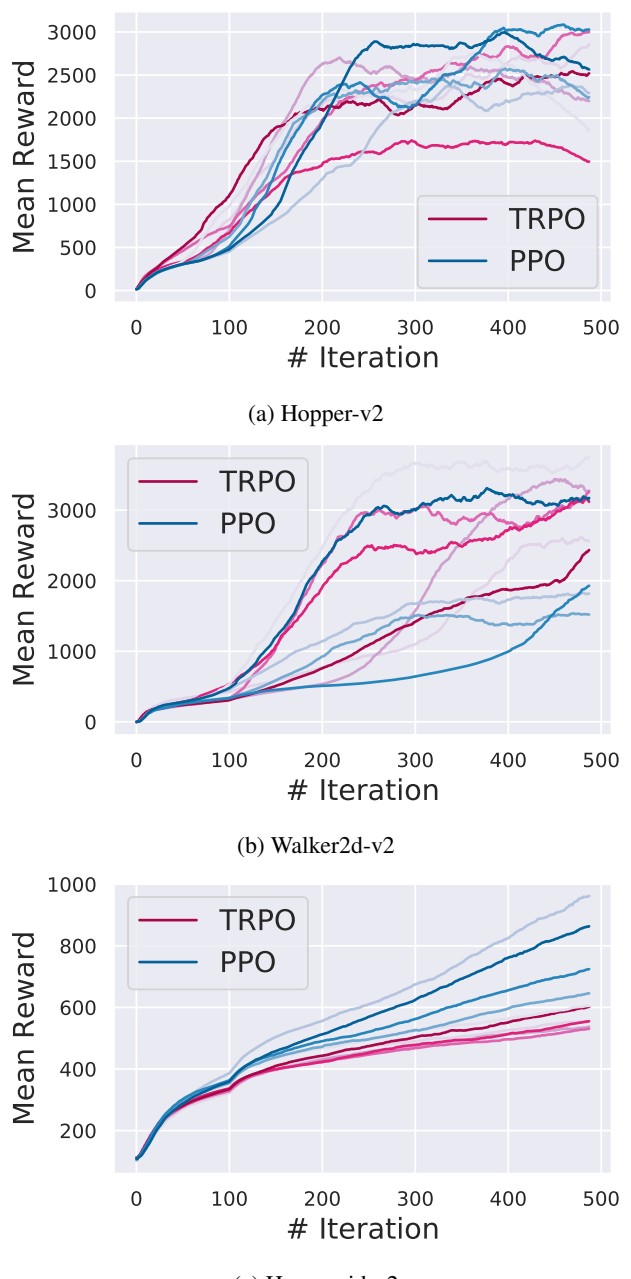

(a) Hopper-v2

(b) Walker2d-v2

(c) Humanoid-v2

Figure 8: Mean reward for the studied policy gradient algorithms on standard MuJoCo benchmark tasks. For each algorithm, we perform 24 random trials using the best performing hyperparameter configuration, with 10 of the random agents shown here.

## A.4 Quality of Gradient Estimation

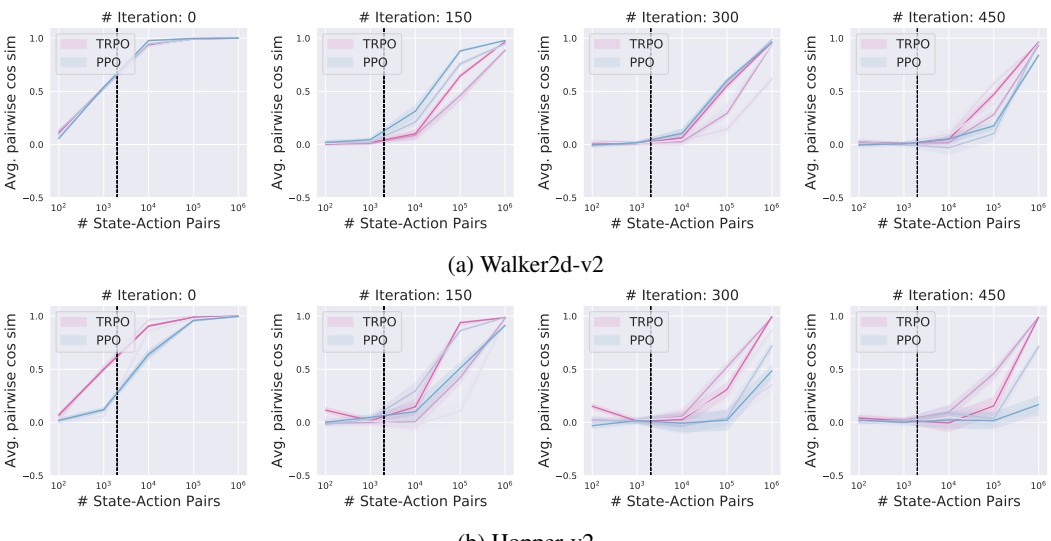

(a) Walker2d-v2

(b) Hopper-v2

Figure 9: Empirical variance of the gradient (c.f. (1)) as a function of the number of state-action pairs used in estimation for policy gradient methods. We obtain multiple gradient estimates using a given number of state-action pairs from the policy at a particular iteration. We then measure the average pairwise cosine similarity between these repeated gradient measurements, along with the $95\%$ confidence intervals (shaded). Each of the colored lines (for a specific algorithm) represents a particular trained agent (we perform multiple trials with the same hyperparameter configurations but different random seeds). The dotted vertical black line (at 2K) indicates the sample regime used for gradient estimation in standard practical implementations of policy gradient methods.

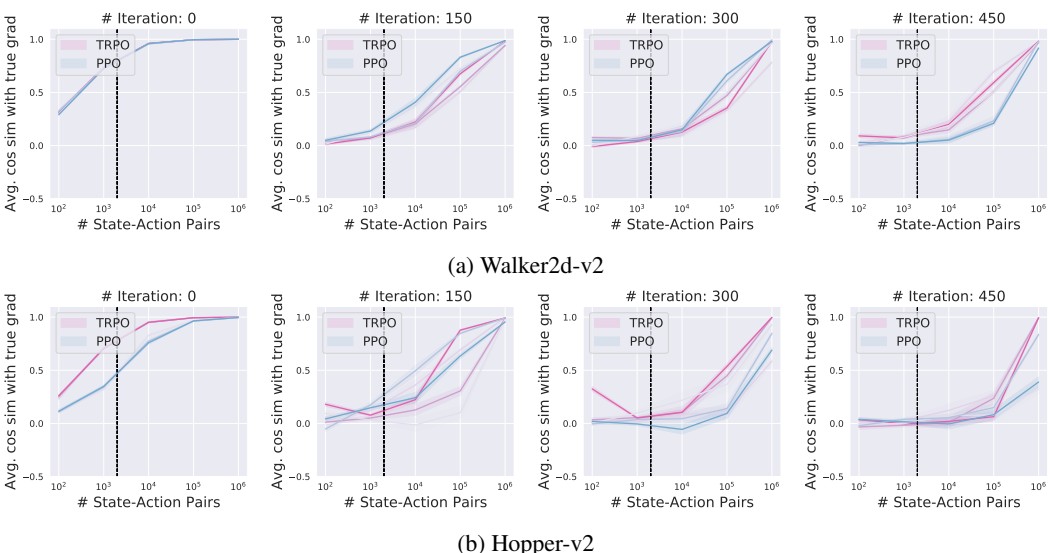

(a) Walker2d-v2

(b) Hopper-v2

Figure 10: Convergence of gradient estimates to the "true" expected gradient (c.f. (1)). We measure the cosine similarity between the true gradient (approximated using around 1M samples) and gradient estimates, as a function of number of state-action pairs used to obtain the later. For a particular policy and state-action pair count, we obtain multiple estimates of this cosine similarity and then report the average, along with the $95\%$ confidence intervals (shaded). Each of the colored lines (for a specific algorithm) represents a particular trained agent (we perform multiple trials with the same hyperparameter configurations but different random seeds). The dotted vertical black line (at 2K) indicates the sample regime used for gradient estimation in standard practical implementations of policy gradient methods.

## A.5 VALUE PREDICTION

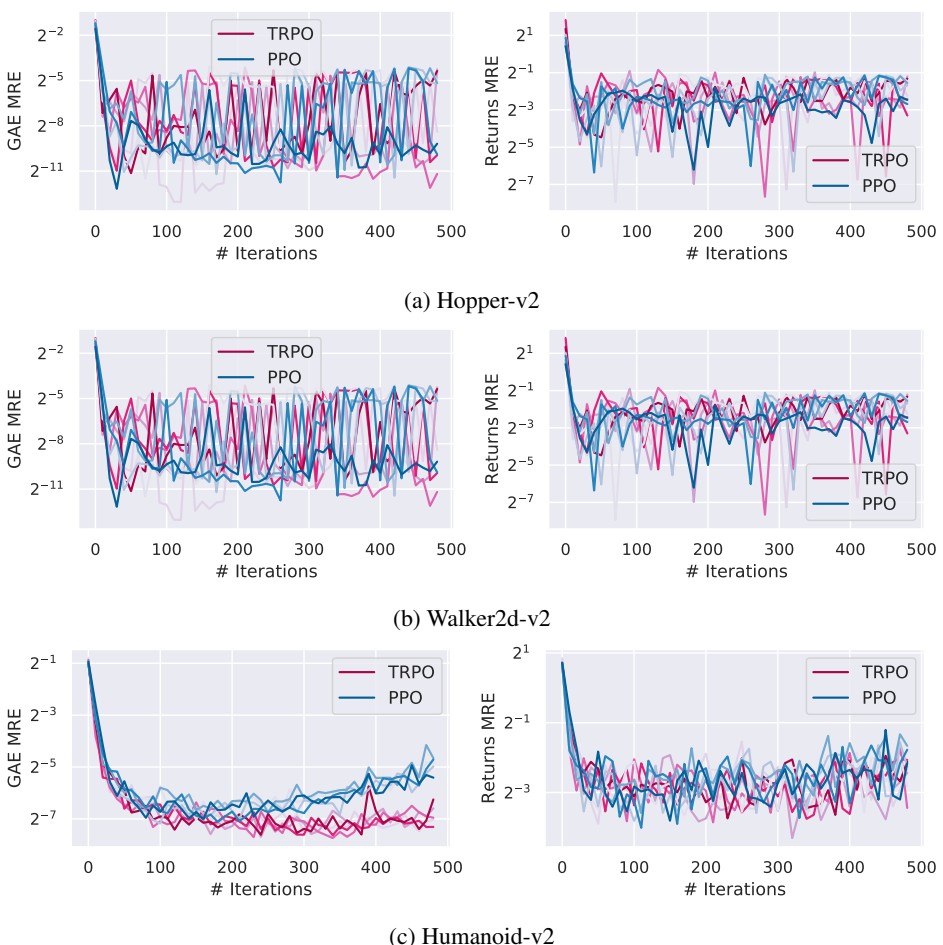

(a) Hopper-v2

(b) Walker2d-v2

(c) Humanoid-v2

Figure 11: Quality of value prediction in terms of mean relative error (MRE) on train state-action pairs for agents trained to solve the MuJoCo tasks. We see in that the agents do indeed succeed at solving the supervised learning task they are trained for – the train MRE on the GAE-based value loss $(V_{old} + A_{GAE})^2$ (c.f. (4)) is small (left column). We observe that the returns MRE is quite small as well (right column).

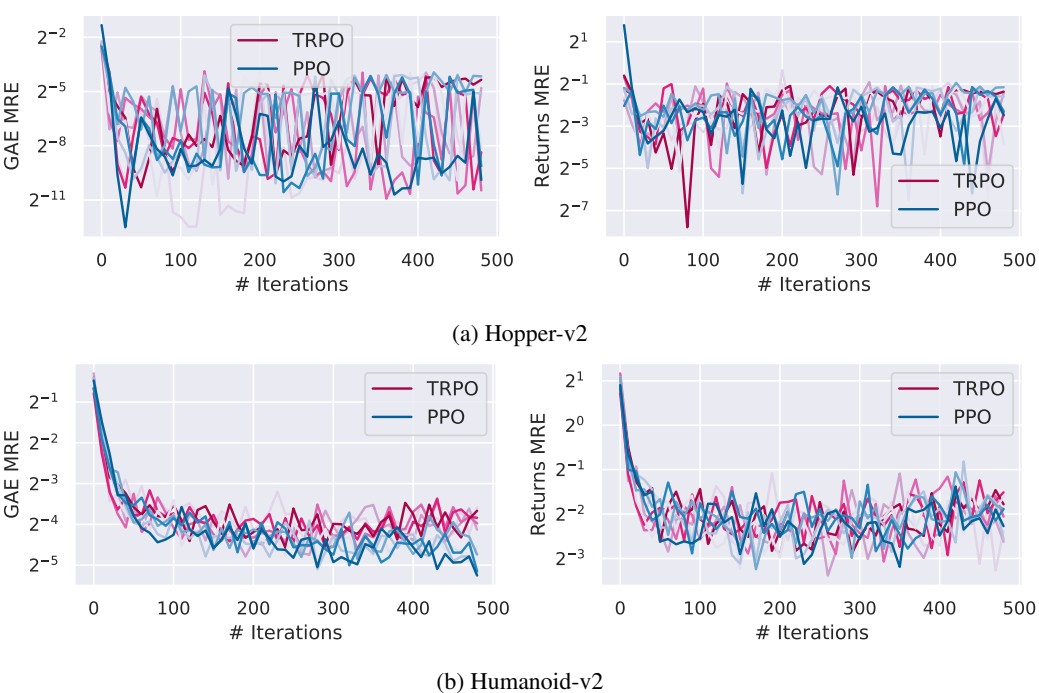

(a) Hopper-v2

(b) Humanoid-v2

Figure 12: Quality of value prediction in terms of mean relative error (MRE) on heldout state-action pairs for agents trained to solve MuJoCo tasks. We see in that the agents do indeed succeed at solving the supervised learning task they are trained for – the validation MRE on the GAE-based value loss $(V_{old} + A_{GAE})^2$ (c.f. (4)) is small (left column). On the other hand, we see that the returns MRE is still quite high – the learned value function is off by about $50\%$ with respect to the underlying true value function (right column).

## A.6 OPTIMIZATION LANDSCAPE

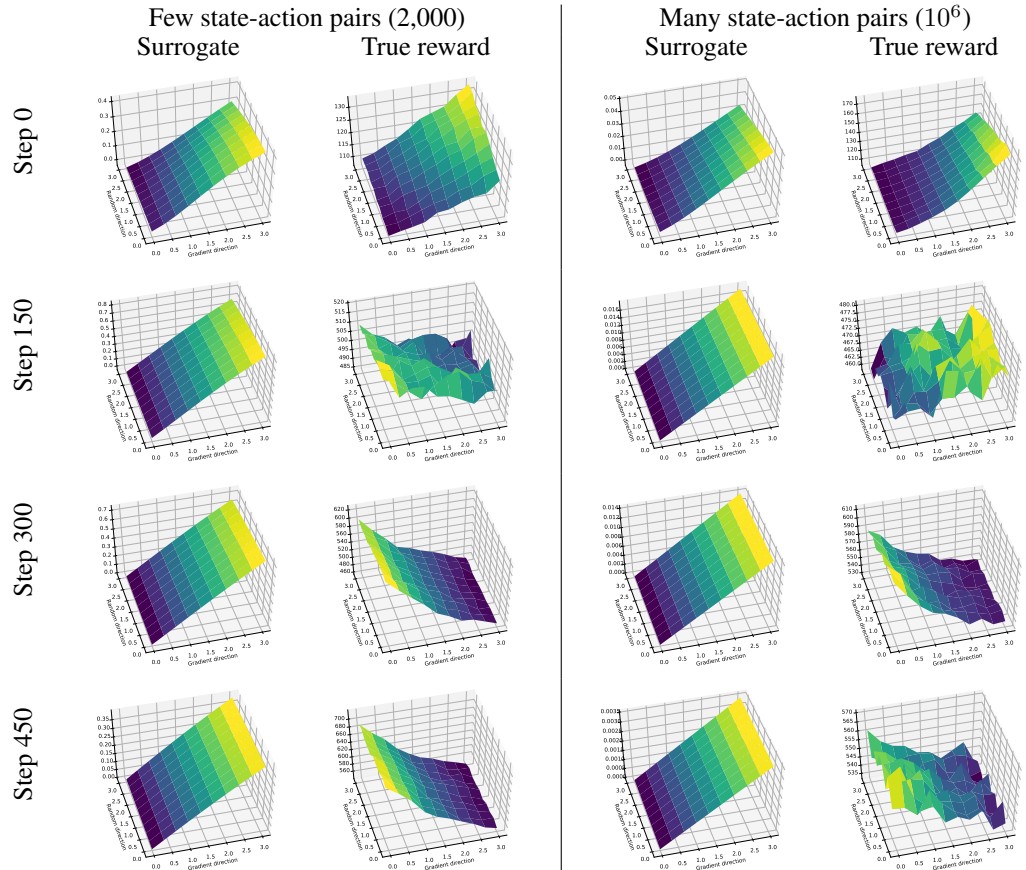

Figure 13: Humanoid-v2 – PPO reward landscapes.

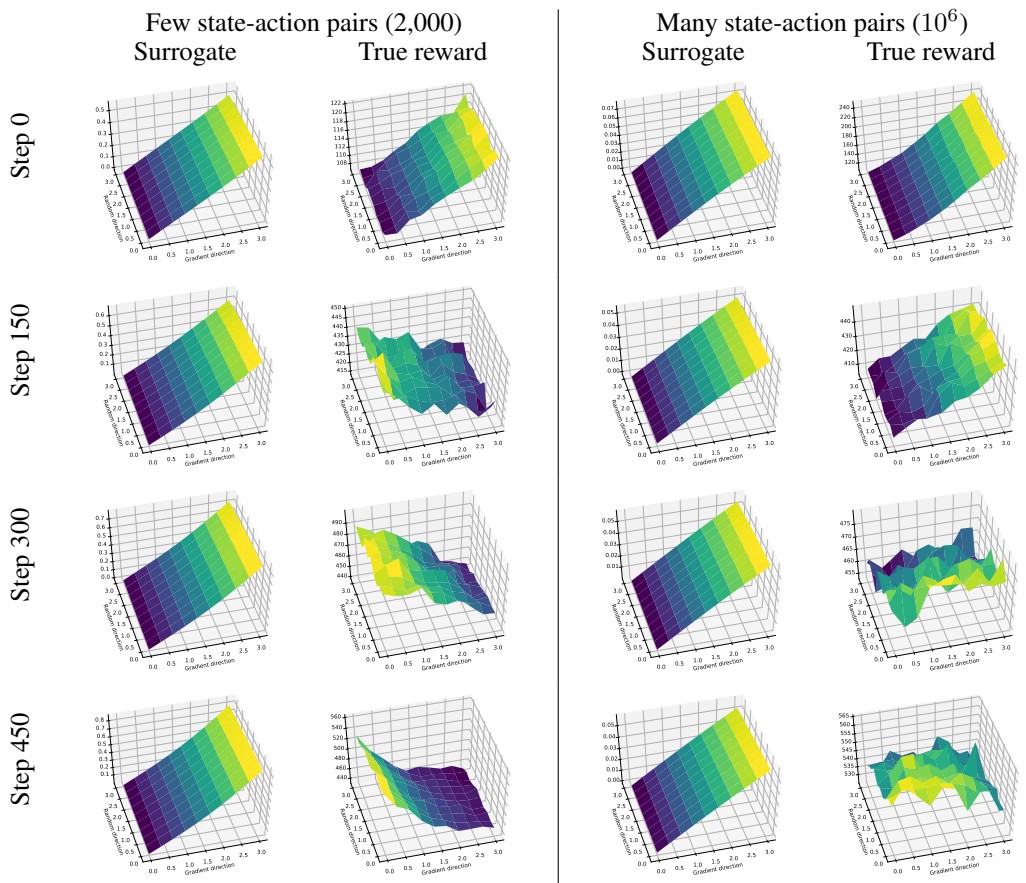

Figure 14: Humanoid-v2 – TRPO reward landscapes.

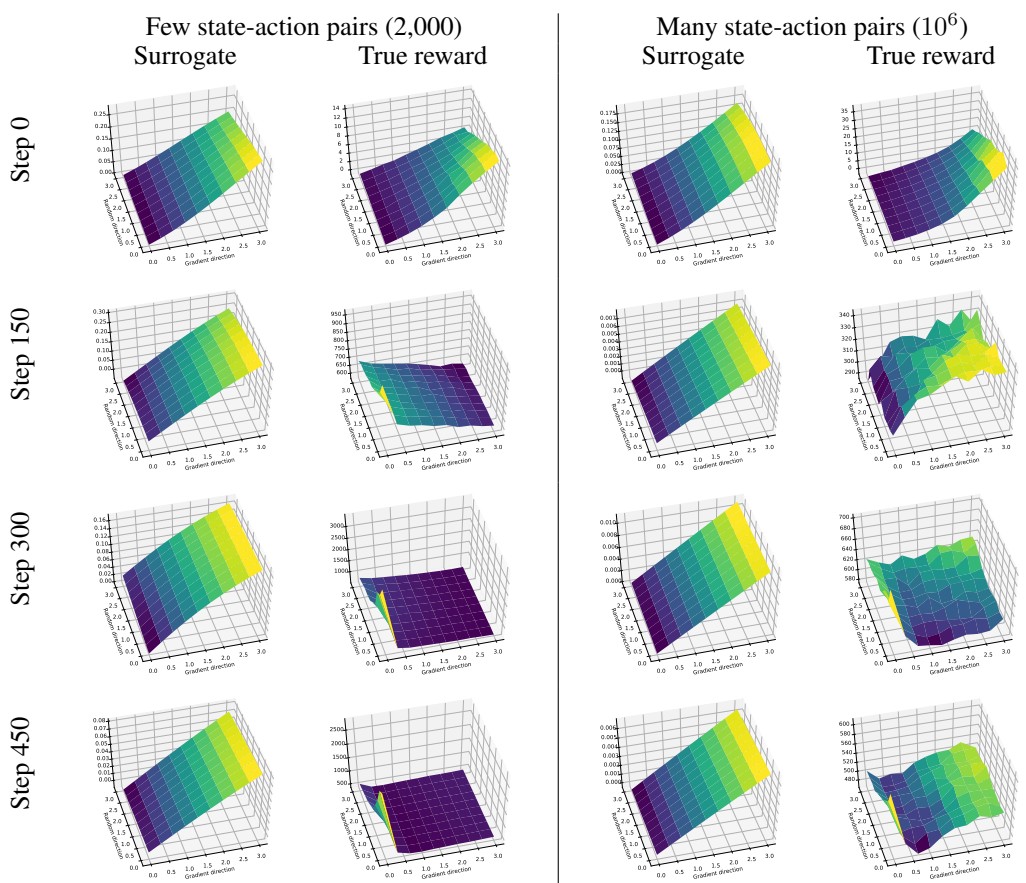

Figure 15: Walker2d-v2 – PPO reward landscapes.

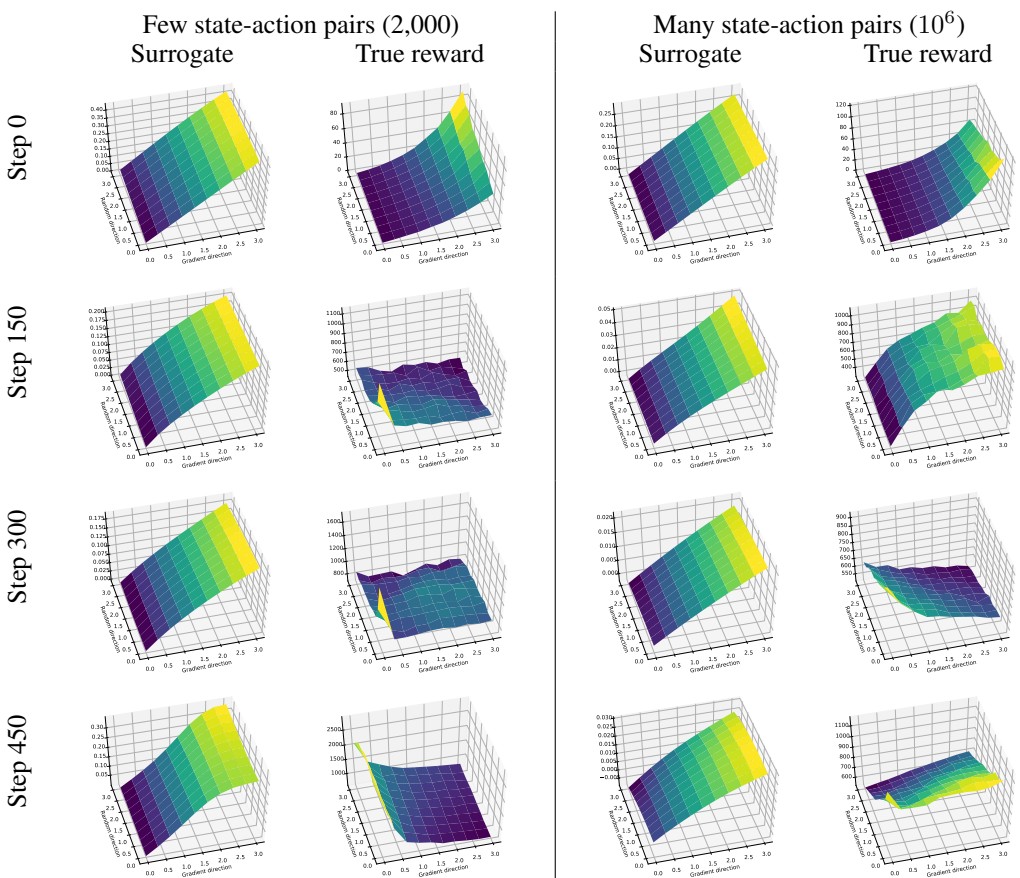

Figure 16: Walker2d-v2 – TRPO reward landscapes.

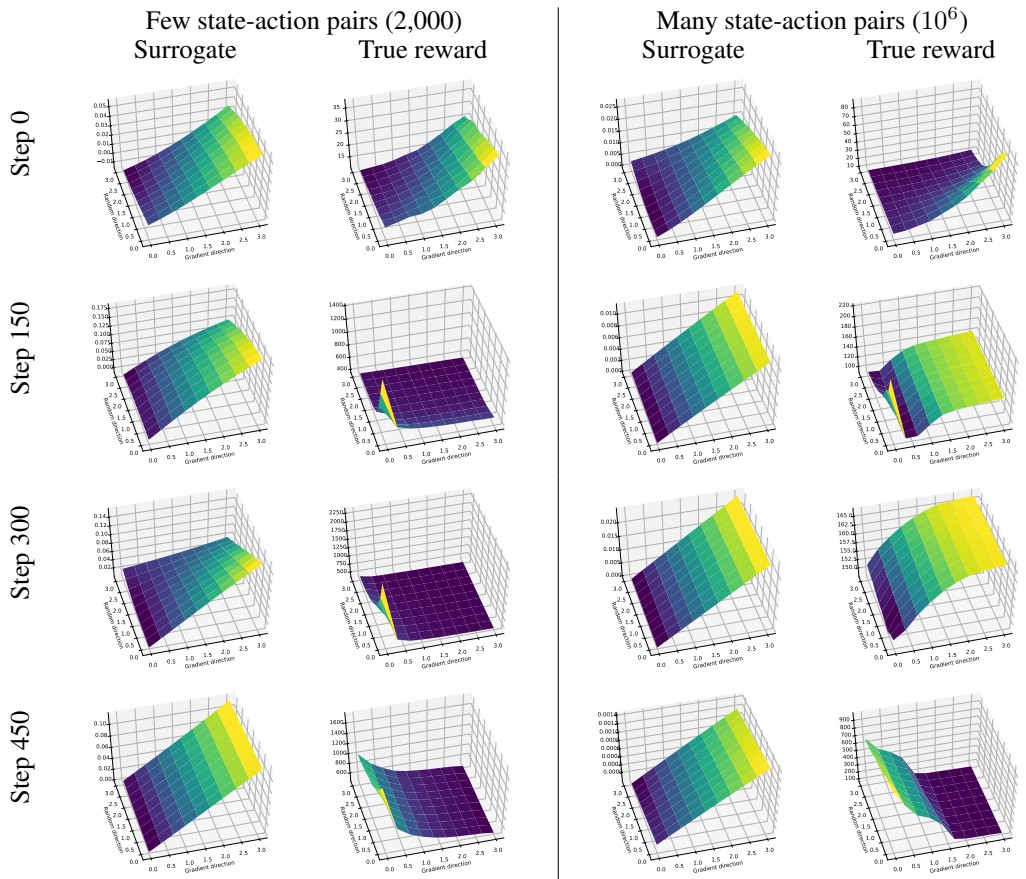

Figure 17: Hopper-v2 – PPO reward landscapes.

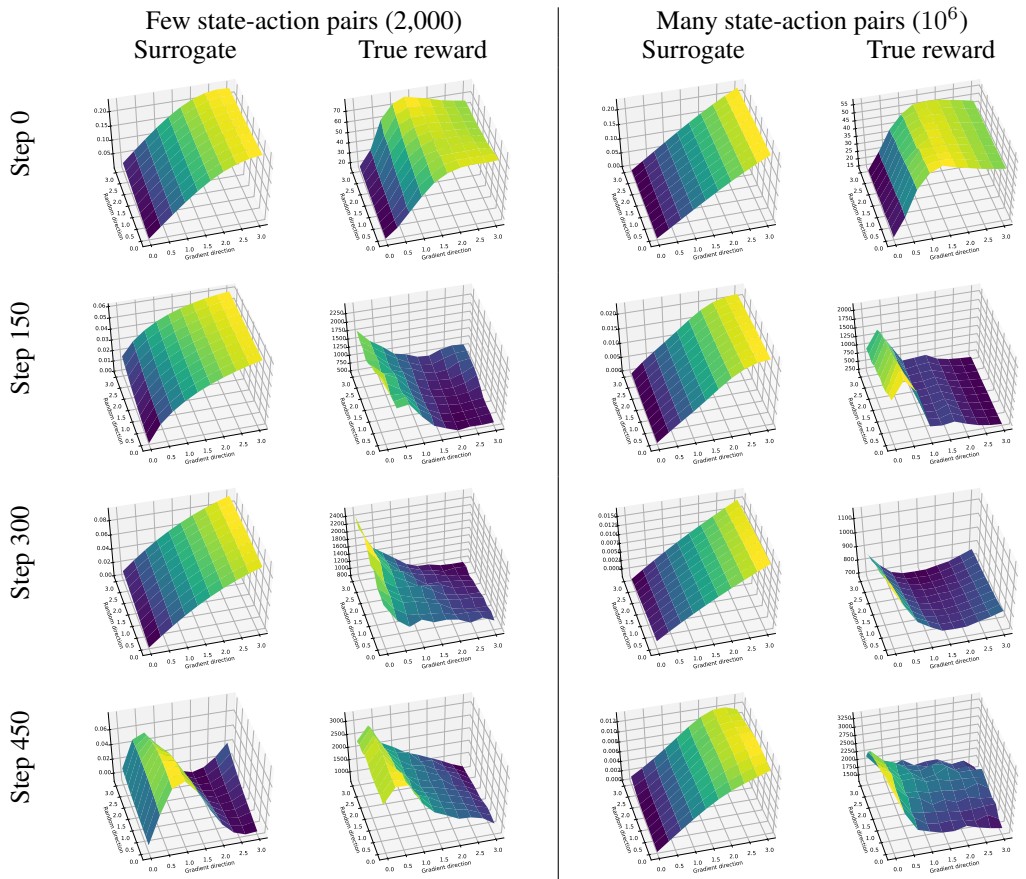

Figure 18: Hopper-v2 – TRPO reward landscapes.

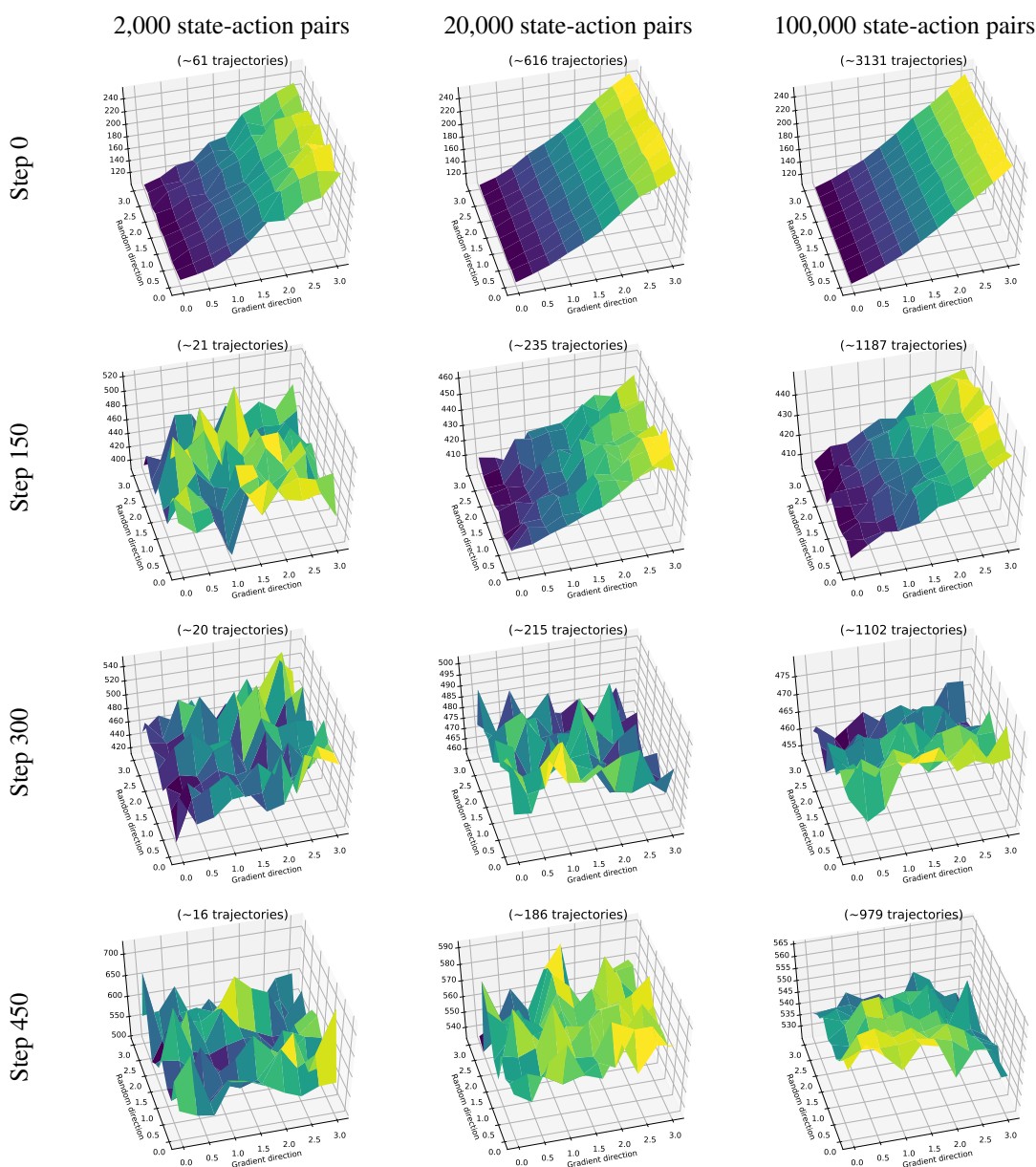

Figure 19: Humanoid-v2 TRPO landscape concentration (see Figure 5 for a description).

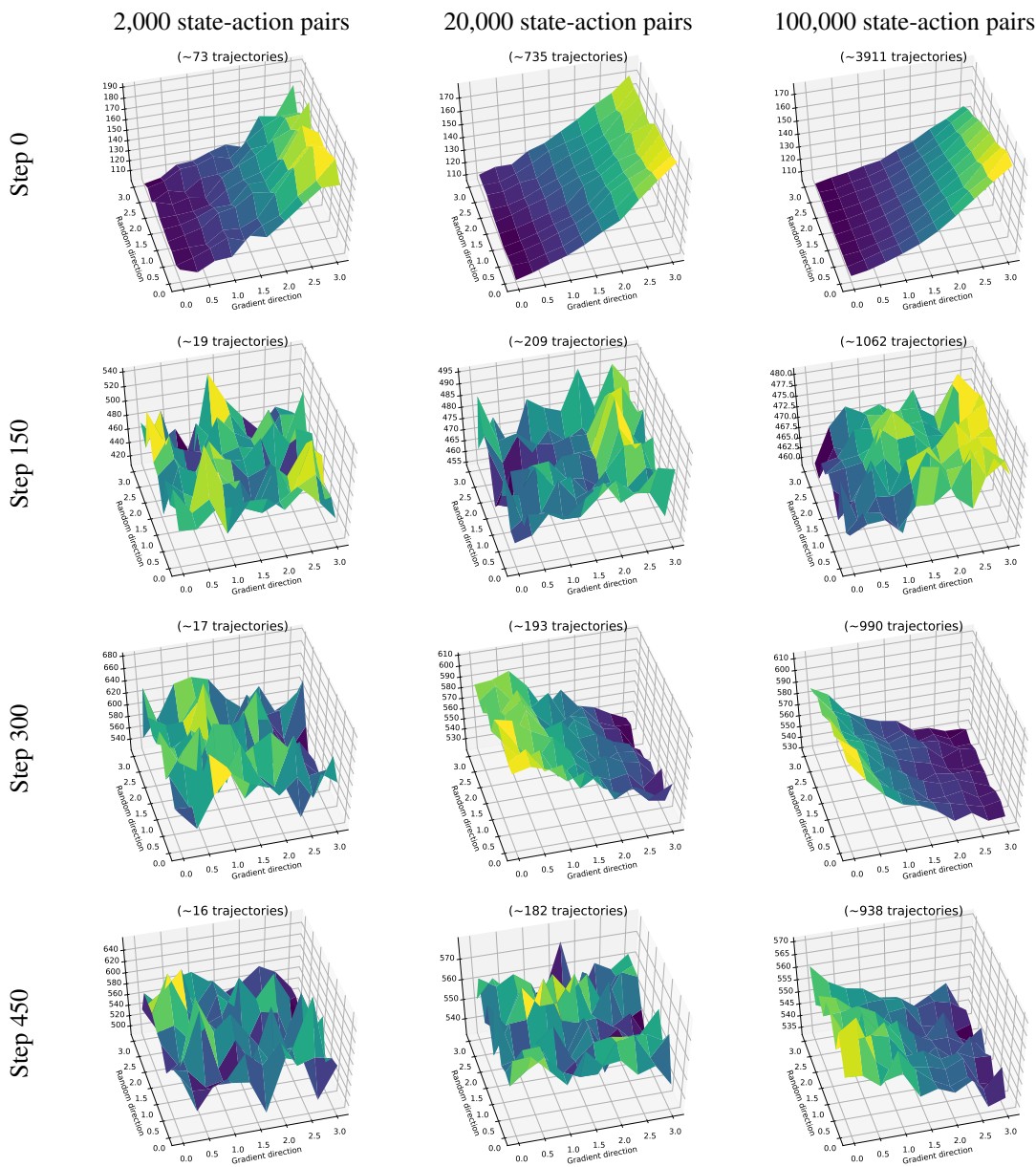

Figure 20: Humanoid-v2 PPO landscape concentration (see Figure 5 for a description).

