# OpenReview forum: "A Closer Look at Deep Policy Gradients"
_ICLR.cc/2020/Conference — Accept (Talk)_

### Official Review · AnonReviewer2 · 2019-10-23
**Official Blind Review #2**

**Rating:** 8

**Review:**

The paper explores a critical divergence between theory and practice, emphasizing that while deep policy gradient algorithms seem to work in certain cases, they don't seem to be working foor the reasons underlying their derivations. It particularly looks at how closely the sample-based approximation of the objective's gradient aligns with the true gradient of the objective, how accurately learned values match the true expected returns, and how well the optimization landscapes of surrogate objectives line up with the objective of maximizing the return.

I propose accepting this paper, as it reveals a key gap in our understanding of why policy gradient methods work. Such emphasis can suggest why deep RL results tend to be inconsistent and irreplicable, and spark future work on closing the gap between theory and practice. Further, the paper is overall well written.

I primarily would like clarification on the optimization landscape visualizations:

1) Is the step direction the direction of the update actually performed at that time step?

2) Would moving diagonally in this space correspond to a mixture of following the update direction and a normally-distributed random direction? Concretely, in the true reward plot at Step 0 for few state-action pairs in Figure 8, does this suggest that mixing a random direction with the update direction would be better than moving cmopletely in the step direction?

Minor:
Typo in citation "...policy improvement theorem of Kakade and Langford Kakade & Langford (2002)"

**Experience Assessment:**

I have published one or two papers in this area.

**Review Assessment: Checking Correctness Of Derivations And Theory:**

N/A

**Review Assessment: Checking Correctness Of Experiments:**

I carefully checked the experiments.

**Review Assessment: Thoroughness In Paper Reading:**

I read the paper thoroughly.

---

> ### Author Response · Authors · 2019-11-13
> **Response**
>
> Thank you for your comments.
>
> Responding to your questions: The step direction is the direction of the update computed for the current agent, and the points do indeed correspond to linearly combined mixtures of a random (Gaussian) vector and the step direction. Your conclusion that following a random direction would be more beneficial than following the step in Figure 8 is correct --- this misalignment between true reward and surrogate objective is a core finding of our work.
>
> We have fixed the citation, thank you!

---

### Official Review · AnonReviewer3 · 2019-10-23
**Official Blind Review #3**

**Rating:** 6

**Review:**


[Summary]
This paper empirically studies the behavior of deep policy gradient algorithms during the optimization. The conclusion is that, while these methods generally improve the policy, their behavior does not comply with the underlying theoretical framework. First, sample gradients obtained with a reasonable batch size have little correlation with each other and with the true gradient. Second, a larger batch size requires a smaller step-size. Third, the value baseline is far from true values and only marginally reduces variance, yet it considerably helps with optimization. Finally, the optimization landscape highly varies with the choice of objective function and the number of samples used to estimate it.

[Decision]
I vote for acceptance. To the best of my knowledge, the findings of this paper are new and not predictable by the current theory. These negative results have some merit as they call for theory that explains the behavior of these algorithms, or an algorithm whose behavior is predictable by the current theory. The paper is well-written, with a few small issues in presentation that should to be addressed in the final revision.

[Comments]
In Fig. 4 (b) it does not look like that the value error is high. It is said that "the learned value function is off by about 50% w.r.t. the underlying true value function." This sentence should be clarified or visualized.

What is \pi in Eq (13) in A1? If it is the agent's current policy, how is it different than \pi_\theta? If \pi corresponds to the distribution of state-action pairs in the replay buffer, how can one obtain a policy \pi that has led to this distribution of states in order to construct the importance sampling ratio?

In 2.2, the claim that a learned value baseline results in significant improvement in performance should be supported by results or reference to previous work.

Figs. 6 and 7 compare the loss surface with different objectives and sample regimes. Do these factors (objective and sample size) affect the part of the parameter space that is visualized (by changing the origin and the update direction), or are they only used to evaluate the values on the z-axis for the same area in the parameter space? Observing a different landscape in a different part of the parameter space is not surprising.

[Minor comments]
- Is V_\theta_{t-1} in Eq (4) a function of state? If so, a (s_t) is missing before the plus sign.

**Experience Assessment:**

I have published one or two papers in this area.

**Review Assessment: Checking Correctness Of Derivations And Theory:**

I carefully checked the derivations and theory.

**Review Assessment: Checking Correctness Of Experiments:**

I carefully checked the experiments.

**Review Assessment: Thoroughness In Paper Reading:**

I read the paper thoroughly.

---

> ### Author Response · Authors · 2019-11-13
> **Response**
>
> Thank you for your comments and suggestions. We have addressed your concerns below:
>
> In Figure 4 (b), the returns MRE hovers around 1/2, which is where we obtain the “50% off of the true value function” conclusion.
>
> In (13), the $\pi$ refers to the current policy, and $\pi_\theta$ refers to the policy we are optimizing to solve the maximization problem (which will become the agent’s new policy).
>
> We have added references to previous work --- Schulman et al 2015 [0] and Sutton and Barto 2018 [1] --- that support the assertion that learned baselines result in significant improvements in agent performance.
>
> In Figure 6 and 7, the changing factor of sample size and objective do not change the way that the agent was run. We ablate these factors to indicate the objective mismatch, and the noisiness of the reward landscape around agents.
>
> Thank you for the catch, the $V_{\theta_{t-1}}$ in Eq (4) is indeed a function of state and we have corrected this notation accordingly.
>
> [0] https://arxiv.org/abs/1506.02438
> [1] http://incompleteideas.net/book/the-book.html

---

### Official Review · AnonReviewer1 · 2019-10-23
**Official Blind Review #1**

**Rating:** 8

**Review:**

This is an interesting and important paper, it emphasizes and analyzes how policy gradient methods modify their objective functions and how this leads to training differences (and often errors w.r.t. the true objective). I have some minor comments on terminology used that I would like to see properly defined within the paper, but otherwise believe this should be accepted for its useful insights.

Assorted Comments:
+ Maybe I simply have a difference of opinion or have misunderstood, but I am hesitant to agree that the work is comparing the surrogate *reward* function, but rather the surrogate objective. You'll notice that in the TRPO paper, it is called a surrogate objective not a surrogate reward: https://arxiv.org/pdf/1502.05477.pdf .
+ I think better specification of what exactly is being plotted (pointing to an equation) or defining very concretely what is a surrogate reward or true reward (which I suspect is the objective) will make this paper much clearer.
+ In fact, it was a bit unclear whether the comparisons were of the sampled/observed reward function R(s,a) (provided by the environment and sampling regime) or the objective function often the advantage A(s,a) (or the surrogate objective, GAE, etc.) I assume it should be the latter, but the wording of the paper makes this a bit unclear. I suggest discussing things in terms of objectives not rewards -- unless in fact the paper does approximate reward functions in which case this should be specified in much more detail.
+ Also, in a lot of places it seems like there's a mixup between rewards and returns. I think typically in the literature reward = r_t and return = V_t (sum of reward). Perhaps, in places the paper truly speaks of rewards, but from the context it seems as though it mainly refers to returns. Examples: " Evidently (since the agent attains a high reward) these estimates are sufficient to consistently improve reward" " This is in spite of the fact that our agents continually improve throughout training, and attain nowhere near the maximum reward possible on each task"


**Experience Assessment:**

I have published in this field for several years.

**Review Assessment: Checking Correctness Of Derivations And Theory:**

I assessed the sensibility of the derivations and theory.

**Review Assessment: Checking Correctness Of Experiments:**

I assessed the sensibility of the experiments.

**Review Assessment: Thoroughness In Paper Reading:**

I read the paper at least twice and used my best judgement in assessing the paper.

---

> ### Author Response · Authors · 2019-11-13
> **Response**
>
> Thank you for your feedback, and we are happy that you enjoyed the paper.
>
> Surrogate objectives/”Surrogate rewards” terminology: we indeed refer to the surrogate objective when we refer to the surrogate reward --- we have corrected this in the revision by replacing all instances of surrogate reward with surrogate objective. The terminology of “surrogate reward” simply refers to the fact that instead of optimizing over the true rewards, agents optimize over a surrogate function. To address your point of concretely defining the surrogate objective, we have placed a reference in the main text to the surrogate objective’s definition (which can be found in the Appendix).
>
> With respect to our experiments/comparisons, our experiments use the surrogate objective or the true reward information depending on the section. We measure steps optimizing the surrogate objective in our gradient estimation quality experiments, and plot both the true reward and the surrogate objective in our landscape experiments.
>
> We agree that it would be an interesting line of work to investigate how the misalignment of the surrogate reward impacts value learning.

---

### Author Response · Authors · 2020-04-02
**Camera Ready Uploaded**

We have updated the paper to the camera-ready version now. While updating it, we saw a bug in our implementation of KL divergence calculation---we have ensured that the results remain accurate by re-running all of the experiments in our paper (the graphs have been updated). Below is a list of the minor edits we made in the camera-ready version:
- While we were rerunning everything, we used many more agents to ensure the trends held up. All of the trends have been verified on 24 random agents.
- We performed a much finer grid search to find the best agent parameters
- We removed the "PPO-M" line from the graphs, as we did not introduce PPO-M in this paper and it was an artifact from an earlier revision.
- We removed the 2-3 sentences about the high-sample regime, since we were unable to reproduce it reliably with the new code/old parameters (and lacked the compute to do another full grid in the high-sample regime)
- We updated the value baseline results to use 5 million state-action pairs instead of 500K
- We give finer detail about the hyperparameters used in Appendix A

---

### Decision · Program_Chairs · 2019-12-19

**Decision:**

Accept (Talk)

**Comment:**

The paper empirically studies the behaviour of deep policy gradient algorithms, and reveals several unexpected observations that are not explained by the current theory. All three reviewers are excited about this work and recommend acceptance.